# Preposition Stranding in Spanish–English Code-Switching

**Bryan Koronkiewicz** 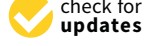

Department of Modern Languages & Classics, The University of Alabama, Tuscaloosa, AL 35487, USA;
bjkoronkiewicz@ua.edu

**Abstract:** This study tests the acceptability of preposition stranding in the intrasentential code-switching of US heritage speakers of Spanish. Because languages vary when extracting determiner phrases from prepositional phrases, known as *preposition stranding* or *p-stranding*, a contrast arises for Spanish–English bilinguals. English allows p-stranding, but in Spanish the preposition is traditionally pied-piped with the DP. Heritage speakers of Spanish, though, have shown variability, with child sequential bilinguals requiring said pied-piping, but simultaneous bilinguals allowing p-stranding in Spanish. Participants (*n* = 24) completed a written acceptability judgment task with a 7-point Likert scale. The task included code-switched sentences (*n* = 16) with p-stranding, switching from either English to Spanish or vice versa, with comparison monolingual equivalents for Spanish (*n* = 8) and English (*n* = 8) included as well. The results found that the simultaneous bilinguals accepted p-stranding in both languages, while also showing no restriction in either code-switching condition. Child sequential bilinguals, however, showed the expected monolingual distinction between Spanish and English, and p-stranding was only accepted with Spanish determiner phrases extracted from an English prepositional phrase (i.e., Spanish-to-English). These findings support the previously reported differentiation between simultaneous and child sequential bilinguals regarding p-stranding, while expanding it to code-switching.

**Keywords:** code-switching; bilingualism; heritage languages; syntax; prepositions; p-stranding; Spanish; English

## 1. Introduction

Languages vary when extracting determiner phrases (DPs) from prepositional phrases (PPs) (Law 2006; Salles 1995). English allows for such extraction, referred to as *preposition stranding* or *p-stranding*, as shown in the sentence in (1). We see that the DP *what friend* originates in sentence-final position as the complement of the preposition *with*. However, the DP does not remain in situ as it is part of an embedded wh-question, which requires the wh-phrase to move to a higher position within the embedded complementizer phrase (CP). This movement consequently 'strands' the preposition at the end of the sentence; the DP has left a trace in this position and has been moved to CP.

Spanish traditionally does not allow for p-stranding. Consider the sentences in (2). We see that, unlike in English, stranding the preposition is generally considered ungrammatical[1] (2a). When wh-movement is required, the preposition is pied-piped along with the wh-element, and as such the entire PP moves as one syntactic unit (2b).

(1)  Chad doesn't know [DP what friend]i Kevin is traveling [PP with ti].

(2)  a.  *  Él    no    sabe    [DP qué    amiga]i  ella    está    viajando    [PP con ti].
            he    not    knows   what     friend   she     is       traveling    with
            'He doesn't know what friend she is traveling with.'

     b.     Él    no    sabe    [PP con   [DP qué    amiga]i   Sergio    está    viajando.
            he    not    knows   with      what     friend     Sergio    is       traveling
            'He doesn't know with what friend she is traveling.'

Given this distinction in p-stranding availability[2], a contrast arises for Spanish–English bilinguals, and it is unclear how this syntactic difference manifests itself when the two languages are mixed. Yet to be tested experimentally is the availability of p-stranding in intrasentential code-switching (CS). If one language permits such a construction while the other rejects it, what happens syntactically when the two languages are combined in the same utterance? In other words, is it possible to extract a Spanish DP out of an English PP (3a) or vice versa (3b)?

(3)   a.   *Fernando*[3]   *no*   *sabe*   [$_{DP}$ *qué*   *amiga*]$_i$   Kevin is traveling [$_{PP}$ with t$_i$].
Fernando   not   knows   what   friend
'Fernando doesn't know what friend Kevin is traveling with.'

b.   Chad doesn't know [$_{DP}$ what friend]$_i$   *Sergio*   *está*   *viajando*   [$_{PP}$ *con* t$_i$].
Sergio   is   traveling   with
'Chad doesn't know what friend Sergio is traveling with.'

The current study provides experimental evidence that p-stranding is available in Spanish–English CS, but it is mitigated by the age of acquisition (AoA) of English of the speaker, as well as the direction of the switch. Specifically, child sequential bilinguals show no restriction whatsoever regarding p-stranding, accepting such a construction in English, Spanish, and CS (in either direction). Simultaneous bilinguals, on the other hand, show an asymmetry between their two languages, accepting p-stranding in English and rejecting it in Spanish; when mixing the two, they only accept it in Spanish-to-English contexts.

## 2. Background

### 2.1. P-Stranding in English and Spanish

Wh-elements in English occupy a higher syntactic position than where they are base generated, generally considered to be the specifier of the CP (Chomsky 1986), as shown in (4a). If the wh-element originates as the complement of a PP, it can be extracted, stranding the preposition in situ in its lower position (Law 2006; Salles 1995), as shown in (4b).

(4)   a.   [$_{DP}$ What]$_i$ did you buy t$_i$?

b.   [$_{DP}$ What money]$_i$ did you buy it [$_{PP}$ with t$_i$]?

In addition to simple wh-movement (4), p-stranding in English can also occur with embedded wh-movement (5), as well as with relative clauses (6).

(5)   I don't know [$_{DP}$ what friend]$_i$ you went shopping [$_{PP}$ with t$_i$].

(6)   Amy is the friend ([$_{DP}$ who]$_i$) I went shopping [$_{PP}$ with t$_i$].

Heavily stigmatized "in the eighteenth century as being colloquial, inelegant, improper, or even harsh, [ . . . ] ever since, end-placed prepositions have been frowned upon in grammar books and usage guides" (Yáñez-Bouza 2006, p. 1). Nonetheless, p-stranding continues to be a common syntactic feature of English (Quirk et al. 1985). It is not the only option, though, as pied-piping can alternatively be used, as shown in (7), being employed more often in formal discourse (Biber et al. 1999).

(7)   a.   I don't know [$_{DP}$ what store]$_i$ she got it [$_{PP}$ from t$_i$].

b.   I don't know [$_{PP}$ from [$_{DP}$ what store]]$_i$ she got it t$_i$.

For most Spanish speakers, p-stranding is disallowed entirely, requiring the preposition to be pied-piped with the DP (Law 2006). Syntactically speaking, this construction directly mirrors the p-stranding alternative available to English speakers, with the crucial distinction being it is the only option. Pied-piping occurs with Spanish simple wh-movement (8), embedded wh-movement (9), and relative clauses (10).

(8) a. * ¿[DP Qué　dinero]　lo　compraste　[PP con t$_i$]?
　　　　What　　money　it　bought.2s　　with
　　　　'What money did you buy it with?'

　　b. ¿[PP Con　[DP qué　dinero]]　lo　compraste t$_i$?
　　　　With　　what　money　it　bought.2s
　　　　'With what money did you buy it?'

(9) a. * No　sé　[DP qué　amiga]$_i$　fuiste　de　compras　[PP con t$_i$].
　　　Not　know.1s　what　friend　went.2s　of　purchases　with
　　　'I don't know what friend you went shopping with.'

　　b. No　sé　[PP con　[DP qué　amiga]]$_i$　fuiste　de　compras t$_i$.
　　　Not　know.1s　with　what　friend　went.2s　of　purchases
　　　'I don't know with what friend you went shopping.'

(10) a. * Irene　es　la　amiga　que　[DP Ø]$_i$　fui　de　compras
　　　Irene　is　the　friend　that　wh-　went.1s　of　purchases
　　　[PP con t$_i$].
　　　with
　　　'Irene is the friend (that/who) I went shopping with.'

　　b. Irene es　la　amiga　[PP con　[DP quien]]$_i$　fui　de　compras t$_i$.
　　　Irene is　the　friend　with　whom　went.1s　of　purchases
　　　'Irene is the friend with whom I went shopping.'

How do we account for p-stranding availability across languages? Specifically, why is it that English has both options available (i.e., p-stranding and pied-piping), whereas Spanish is limited to just the one (i.e., pied-piping)? Law (2006) proposes that the availability is "related to the independent grammatical property of [the determiner] incorporating into [the preposition]" (p. 633). He argues that languages like Spanish (including other Romance languages and Germanic languages except English) are subject to a syntax-morphology-interface condition where "elements that undergo suppletive rules must form a syntactic unit X°" (Law 2006, p. 647). This condition is evidentially based on suppletive forms such as *de el = del* 'of the' and *a el = al* 'to the' in Spanish, as well as *a les = aux* 'to the' in French, *con lo = collo* 'with the' in Italian, *an dem = am* 'at the' in German, and so on. Note that it is the incorporation that allows for the suppletion, not the other way around. As Law argues, this syntax-morphology-interface condition does not always require suppletion for the syntactic unit to be formed; if a language has such forms at all, it is evidence that all determiners incorporate, with or without suppletion. As such, although Spanish suppletion is not robust, consisting solely of *del* 'of the' and *al* 'at the', Law argues that all Spanish D+P constructions (e.g., *de la* 'of the', *a la* 'at the', *con el* 'with the', etc.) involve incorporation and must syntactically operate as one unit. For wh-movement, for example, the only option then is to move the entire Spanish PP, as the wh-element has formed a syntactic unit with the preposition, as shown in (11).

(11) Araceli　no　sabe　[PP [P+D con qué$_i$]]　[DP t$_i$ hombre]]$_j$　Rosario
　　Araceli　not　know.3s　with-what　　man　　　Rosario
　　está　comiendo t$_j$.
　　is　eating
　　'Araceli doesn't know with what guy Rosario is eating.'

English, lacking any such suppletive forms, does not have the same syntax-morphology interface condition of D+P incorporation. Therefore, in English wh-movement, for example, the wh-element can be freely extracted since it can be syntactically separated from the preposition, as shown in (12a); additionally, this lack of incorporation does not prevent optional pied-piping, as shown in (12b).

(12) a. Emma doesn't know [DP what guy]$_i$ Jen is eating [PP [P with] t$_i$].

　　b. Emma doesn't know [PP [P with [DP what guy]]]$_i$ Jen is eating t$_i$.

### 2.2. Heritage Speaker Bilingualism

A heritage language can be defined as "a language spoken at home or otherwise readily available to young children ... [that] is not a dominant language of the larger (national) society" (Rothman 2009, p. 156). Although heritage grammars and monolingual grammars are similar in that they are acquired via naturalistic input at a young age, substantial differences have been found between them (e.g., Montrul 2008; Silva-Corvalán 1994; among others). Numerous linguistic phenomena have been found to manifest differently in the grammars of heritage speakers, with specific examples from Spanish including subject and object expression (Montrul 2004), subject-verb inversion (Cuza 2013), and presentational focus (Hoot 2017), to name a few.

It is important to note that heritage speakers, "unlike mature monolingual speakers who ultimately converge into a common steady state, exist in a linguistic continuum with no clear or easily measurable cut-off points" (Pascual y Cabo and Soler 2015, p. 188). Although any group of speakers is heterogeneous by nature, acquiring two languages instead of one exponentially increases the potential for differences among the language backgrounds, meaning that a meaningful degree of variation among such bilinguals (even from the same speech community) is not merely a possibility but an inevitability.

Age of acquisition (AoA) is just one of many variables that has been used to better understand within-group differences for heritage language speakers. Essentially, researchers can look at whether the point at which an individual is exposed to the dominant societal language results in different linguistic outcomes for a given grammatical property (Arnaus Gil et al. 2021; Schulz and Grimm 2019; Tsimpli 2014). Although the specific age range varies throughout the literature, the term *simultaneous bilingual* is commonly used for an individual who is exposed to both languages from birth (or a very early age) while the term *sequential bilingual* refers to someone "who first acquired one language, then a second language *after* the rudiments of the first language were established" (Montrul 2008, p. 94). This classification is just a starting point, as the term sequential bilingual can encompass a wide variety of individuals. As such, some researchers have made even further distinctions, for example, delineating *early* sequential bilinguals (AoA 5;0) and *child* sequential bilinguals (AoA 3;0) (Meisel 2013; Montrul 2008). For the current study and its focus on US heritage speakers of Spanish, the AoA of English is used to determine if an individual is either a simultaneous or child sequential bilingual, as that is the language where age of exposure will vary more considerably.

### 2.3. P-Stranding in Heritage Speaker Spanish

One specific syntactic phenomenon that has been shown to diverge between monolinguals and heritage speakers of Spanish is p-stranding. Interestingly, Depiante and Thompson (2013) found evidence that US heritage speakers consistently rated p-stranding as more acceptable in Spanish than a Spanish-dominant control group of recent immigrants. Given this data, it would suggest that there would be no potential conflict for p-stranding in Spanish–English code-switching for these individuals. However, Pascual y Cabo and Soler (2015) found evidence that "the English structure bleeds into [the] Spanish" (p. 203) of some, but importantly not all, heritage speakers. In their study, they targeted *con* 'with' and *en* 'in' in simple wh-questions, embedded wh-questions, and relative clauses across three experimental tasks: (a) a judgment task with p-stranding in Spanish sentences (*n* = 30), (b) a judgment task with pied-piping in Spanish sentences (*n* = 30), and (c) a production task with 'dehydrated' Spanish sentences[4] (*n* = 10). They included two different experimental groups that differed according to US heritage speakers' AoA of English. The first group comprised bilingual speakers (*n* = 21) who learned both Spanish and English from birth, whereas the second group comprised child sequential bilinguals (*n* = 12) who did not learn English until after age 6. Both groups had intermediate/advanced proficiency in Spanish, as well as comparable self-rated proficiencies. A comparison group of Spanish native speakers (*n* = 11) born and raised in Mexico until at least the age of 16 also participated in the study.

The findings show, first, that p-stranding in Spanish was rejected by the comparison monolingual group, whereas it was possible for (some) heritage speakers. Crucially, the two heritage speaker groups diverged, with child sequential bilinguals exhibiting the expected rejection of p-stranding in Spanish, which mirrored the comparison group of monolingual speakers. The simultaneous bilinguals, however, showed no restriction for p-stranding in Spanish, suggesting that the construction has been extended from English for these individuals, as it is "a domain of grammar vulnerable to crosslinguistic influence during the formative years" (Pascual y Cabo and Soler 2015, p. 203). In short, when it comes to p-stranding in heritage speaker Spanish, there is evidence of a categorical difference based on AoA of English.

### 2.4. Research Questions and Hypotheses

The current study adopts a constraint-free approach to CS, and it is agnostic to whether it needs to be a traditional minimalist approach to CS that is lexicalist (MacSwan 1999, 2014, 2021) or exoskeletal (Grimstad et al. 2018). Both are within the generative minimalist program (Chomsky 1995), but differ regarding whether syntactic structures are generated from features on lexical items (i.e., lexicalist) or generated by features independently of lexical items (i.e., exoskeletal). Under such an approach, constraints are due to the interaction of the two grammars in question, specifically when there is a mismatching of features. This mismatching mirrors exactly what happens in monolingual derivations; in other words, this approach assumes there is no 'third grammar' that is unique to CS.

Within this framework, the following research question and sub-question are formulated:

(13) Do heritage speakers of Spanish accept p-stranding in Spanish–English CS, and if so, does the AoA of English play a role?

Based on previous research, we can formulate specific hypotheses to these questions: yes, heritage speakers of Spanish should find p-stranding acceptable in CS; and yes, the results should vary based on the age of onset of bilingualism. Assuming the simultaneous bilinguals replicate the findings of Pascual y Cabo and Soler (2015) and accept p-stranding in Spanish, and assuming the construction is syntactically parallel across the two grammars (i.e., they have English-like non-incorporation of D+P across the board), there should be no restriction whatsoever against p-stranding in CS for these speakers. If they allow p-stranding monolingually in both of their languages, following Law's (2006) analysis, the DP can be freely extracted as an independent syntactic unit (regardless of whether an English DP is extracted from a Spanish PP or vice versa). However, the same cannot be said for the child sequential bilinguals. Assuming they reject p-stranding in Spanish, this would be evidence of D+P incorporation, thus exposing an asymmetry between their Spanish and English grammars. Consequently, there should also be a restriction on p-stranding in CS, with three different possibilities. The child sequential bilingual CS results will depend on which element(s) motivate(s) D+P incorporation: the determiner, the preposition, or both. Specifically, the three options are: (a) if incorporation is dependent only upon the features inherent to the determiner, p-stranding should be only accepted with English-to-Spanish switches (i.e., an English DP with a Spanish preposition); (b) if incorporation is dependent only upon the features inherent to the preposition, p-stranding should only be accepted with Spanish-to-English switches (i.e., a Spanish DP with an English preposition); or (c) if incorporation is dependent upon the features of both the determiner and the preposition, then p-stranding should be rejected in all switch cases.

To summarize, simultaneous bilinguals are hypothesized to accept p-stranding in monolingual English and Spanish, as well as both CS scenarios (i.e., Spanish-to-English and English-to-Spanish). Child sequential bilinguals, however, are hypothesized to reject p-stranding in monolingual Spanish, accept it in monolingual English, and at least some (if not all) p-stranding in CS will be rejected.

### 3. Methods

*3.1. Procedure*

The central component of the experiment was a written[5] acceptability judgment task (AJT) that used a 7-point Likert scale. Each sentence was presented one at a time with the question *How acceptable is this sentence?* or *¿Qué le parece esta oración?* 'How is this sentence to you?'. The points on the scale were labeled as follows: 1 = *completely unacceptable/completamente inaceptable*, 2 = *mostly unacceptable/mayormente inaceptable*, 3 = *somewhat unacceptable/un poco inaceptable*, 4 = *unsure/no sé*, 5 = *somewhat acceptable/un poco aceptable*, 6 = *mostly acceptable/mayormente aceptable*, 7 = *completely acceptable/completamente aceptable*. While the monolingual blocks used their respective questions and labels, the CS block used the Spanish question and English labels in an effort to enhance bilingual mode.

Completed entirely online via the software Qualtrics (Qualtrics, LLC, Provo, UT, USA), participants saw a sentence, clicked their response on the scale, and then clicked an arrow button in the lower right portion of the screen to advance to the next stimulus. The AJT was untimed, meaning participants could take as much time as they liked to complete these steps for each item, and the stimuli were completely randomized within each block for every participant. An example of what the AJT looked like for the participants is shown in Figure 1.

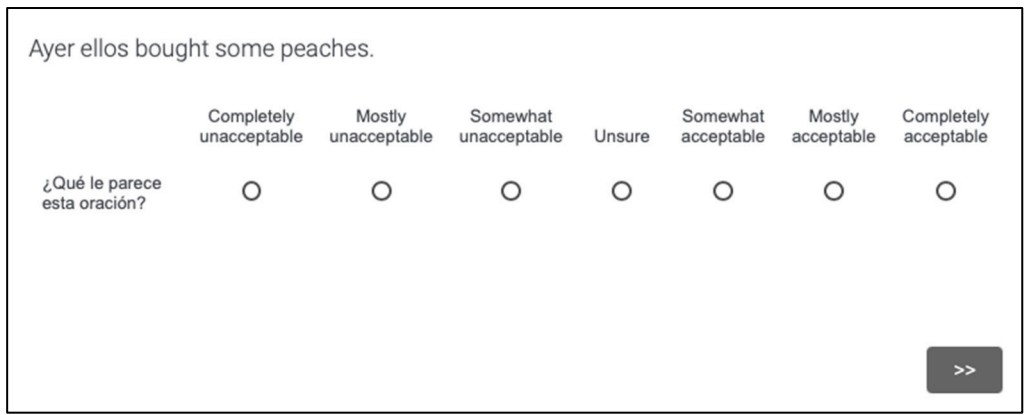

**Figure 1.** Example stimulus from the CS block of the AJT.

The AJT was embedded within a larger experimental procedure that included five sections. First, participants completed a consent form and a brief task training. The training included a general overview of what AJTs ask an individual to do (i.e., use their intuitions, avoid prescriptive rules, etc.) while also including several practice sentences. Following González-Vilbazo et al. (2013), this first section also primed participants for bilingual mode via its use of Spanish–English CS throughout the instructions (with all switches occurring at the clausal boundary or with adjunct phrases). Next, participants completed an entire block of CS judgments. The third section of the procedure switched to monolingual Spanish mode, with a multiple-choice cloze test proficiency measure (Montrul and Slabakova 2003) and then an entire block of Spanish judgments[6]. Afterward, the participants switched to monolingual English mode, which included a different multiple-choice cloze test proficiency measure (O'Neill et al. 1981) followed by a block of monolingual English judgments. Finally, participants wrapped up the procedure with a background questionnaire.

*3.2. Participants*

The individuals who participated in the study were all US heritage speakers of Spanish (*N* = 29), categorized as such by virtue of being second-generation immigrants whose home language was a minority language (i.e., Spanish), as well as being more fluent in the dominant societal language (i.e., English) (Polinsky and Kagan 2007; Valdés 2000). Some

participants were removed from the dataset (*n* = 5) for either not being a self-reported code-switcher and/or for indicating a negative attitude towards CS (Badiola et al. 2018). The remaining participants (*n* = 24) were between 19 and 49 years old (*M* = 23.2) and were either born in the US (*n* = 20) or arrived as a young child (*M* = 4.8 years). Importantly, all participants learned both Spanish and English from a young age.

Following Pascual y Cabo and Soler (2015), the participants were divided into two different groups based on their AoA of English. The simultaneous heritage speakers reported learning both languages at about 1 year of age, with the cut-off being before age 5 (*n* = 13), whereas the sequential heritage speakers learned English at age 5 or later (*n* = 11). This division in the participants was chosen to align with Pascual y Cabo and Soler (2015), thereby making the results directly comparable. An overview and comparison of the two groups' general linguistic profiles is provided in Table 1.

**Table 1.** Participant overview.

| Factor | Simultaneous Bilinguals | | Child Sequential Bilinguals | |
|---|---|---|---|---|
| | *M* | *SD* | *M* | *SD* |
| Age of acquisition: | | | | |
| English | 1.3 | 1.6 | 5.5 | 1.5 |
| Spanish | 0.3 | 0.5 | 0.3 | 0.5 |
| Proficiency score [1]: | | | | |
| English (out of 40) | 37.1 | 1.9 | 36.4 | 2.0 |
| Spanish (out of 50) | 36.9 | 6.2 | 37.6 | 4.7 |
| Self-rated proficiency [2]: | | | | |
| English (out of 5) | 5.0 | 0.0 | 4.9 | 0.3 |
| Spanish (out of 5) | 4.5 | 0.5 | 4.5 | 0.7 |
| English dominance [3] (out of ±218) | 31.3 | 43.0 | 20.0 | 25.1 |

[1] Proficiency was measured via two separate multiple-choice cloze tests for English (O'Neill et al. 1981) and Spanish (Montrul and Slabakova 2003). [2] The scale for self-rated proficiency was as follows: 1—*Poor*; 2—*Needs work*; 3—*Good*; 4—*Very good*; 5—*Native speaker command*. [3] Language dominance was measured via the Bilingual Language Profile (Birdsong et al. 2012).

*3.3. Stimuli*

All target sentences included in the AJT contained p-stranding (*N* = 32). Half of the stimuli included embedded wh-movement and half included relative clauses. There were 8 tokens for each of the 4 language conditions: monolingual English, monolingual Spanish, English-to-Spanish CS, and Spanish-to-English CS. Examples of the embedded wh-movement p-stranding stimuli for each language condition are presented in (14), and examples of the relative clause p-stranding stimuli are in (15).

(14)  a.  Bill doesn't know what woman Megan is arguing with.

  b.  Manuel    no      sabe      qué      señora    Ximena    está      discutiendo
      Manuel    not     know.3s   what     lady      Ximena    is        arguing
      con.
      with
      'Manuel doesn't know what lady Megan is arguing with.'

  c.  Bill doesn't know what woman          *Ximena*    *está*      *discutiendo*    *con.*
                                            Ximena      is          arguing          with
      'Bill doesn't know what woman Ximena is arguing with.'

  d.  *Manuel*    *no*    *sabe*    *qué*    *señora*    Megan is arguing with.
      Manuel      not     know.3s   what     lady
      'Manuel doesn't know what lady Megan is arguing with.'

(15)  a.  Lucy is the girl that Gabe is going out with.

b.  Leticia es la chica que Arturo está saliendo con.
Leticia is the girl that Arturo is going-out with
'Leticia is the girl that Arturo is going out with.'

c.  Lucy is the girl *que Arturo está saliendo con.*
that Arturo is going out with
'Lucy is the girl that Arthur is going out with.'

d.  *Leticia es la chica* that Gabe is going out with.
Leticia is the girl
'Leticia is the girl that Gabe is going out with.'

Filler stimuli with various other types of constructions (and switches for the CS conditions) were also included (*N* = 169). These sentences primarily targeted adverb order, auxiliary verbs, and pronouns.[7] Examples of monolingual filler stimuli are shown in (16–17), and examples of CS filler stimuli are shown in (18–19). The full set of stimuli included in the experiment can be found in Appendix A.

(16) a. *   Sarah buys frequently tomatoes.

b.  Sarah frequently buys tomatoes.

(17) a.  Margarita compra frecuentemente tomates.
Margarita buys frequently tomatoes
'Margarita frequently buys tomatoes.'

b.  Margarita frecuentemente compra tomates.
Margarita frequently buys tomatoes
'Margarita frequently buys tomatoes.'

(18) a. *  *Sus hermanos han* eaten pizza every day.
Their brothers have
'Their brothers have eaten pizza every day.'

(17) b.  *Sus hermanos* have eaten pizza every day.
Their brothers
'Their brothers have eaten pizza every day.'

(19) a. *  *Ayer ellos* bought some peaches.
Yesterday they
'Yesterday they bought some peaches.'

b.  *Ayer esos hombres* bought some peaches.
yesterday those men
'Yesterday those guys bought some peaches.'

## 4. Results

The acceptability scores provided by participants were first standardized, converting them into z-scores. The mean ratings are presented in Figure 2[8], separated out by the two participant groups and four different language conditions. Recall that the higher end of the scale was labeled "acceptable," and as such a more positive z-score is indicative of a sentence's increased acceptability. Note, though, that there is no finite point on the *y*-axis that we can label as "acceptable" or "unacceptable"; what we are interested in rather is comparing which structures were found to be more acceptable or less acceptable relative to each other.

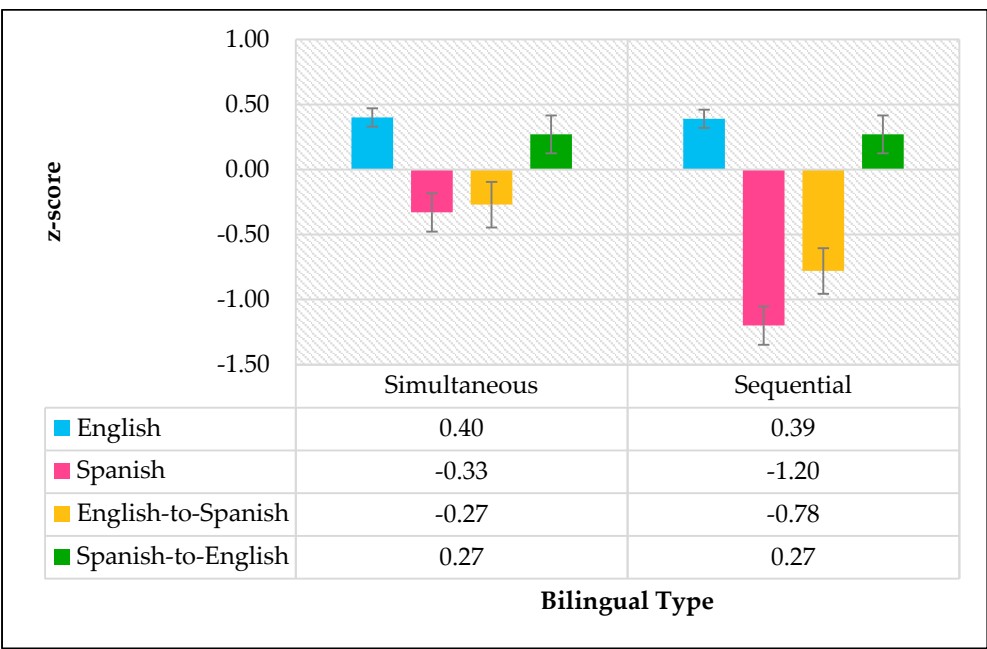

**Figure 2.** Average z-score by language(s) and bilingual type.

If we look first at just the ratings for the monolingual stimuli, we see the expected pattern. The results for the child sequential bilinguals clearly show the crosslinguistic asymmetry, with much higher acceptability ratings for p-stranding in English ($M = 0.39$, $SD = 0.60$) when compared to Spanish ($M = -1.20$, $SD = 0.82$). The results for the simultaneous bilinguals are similar in that they also show a preference for p-stranding in English ($M = 0.40$, $SD = 0.44$), with nearly identical ratings to those of the child sequential bilinguals; however, the two groups diverge in that the simultaneous bilinguals did not seem to outright reject p-stranding in Spanish in that they rated it much higher than the child sequential bilinguals, with scores hovering much closer to the middle of the scale ($M = -0.33$, mboxemph$SD = 0.93$).

Turning to the CS results, the availability of p-stranding in mixed sentences is essentially a mirror image of the monolingual results. The crosslinguistic asymmetry of p-stranding for child sequential bilinguals carries over to the code-switched stimuli, as they provided much lower acceptability scores for p-stranding in English-to-Spanish CS ($M = -0.78$, $SD = 0.87$), which as a reminder is exemplified in (14c) and (15c), as compared to Spanish-to-English ($M = 0.27$, $SD = 0.94$), as exemplified in (14d) and (15d). The simultaneous bilinguals similarly preferred p-stranding when the switch occurred from Spanish-to-English ($M = 0.27$, $SD = 0.92$); however, like in the monolingual results, they did not outright reject p-stranding in English-to-Spanish switches like the other group as their ratings were much higher ($M = -0.27$, $SD = 1.11$).

A two-way ANOVA was conducted to investigate the effect of bilingual type and language(s) on z-score, and a significant interaction was found, $F(3,375) = 7.777$, $p < 0.001$. Post hoc analysis revealed that both groups showed a significant asymmetry for the monolingual stimuli, rating p-stranding in English more favorably than in Spanish ($p < 0.001$). Importantly, though, the groups also differed from each other in that the simultaneous bilinguals were overall more accepting of p-stranding than child sequential bilinguals, regardless of the language ($p < 0.001$). Post hoc analysis also confirmed the parallel nature of the monolingual and CS results; that is, there was no significant difference found between the English and Spanish-to-English ratings ($p > 0.005$), nor was there a significant difference between the Spanish and the English-to-Spanish ratings ($p > 0.05$).

Before concluding the results section, it is worthwhile to explore AoA a bit more in detail. Although the expected difference was found between the two groups of heritage speakers of Spanish, recall that the simultaneous bilinguals did not accept p-stranding to the

same degree in both of their languages, as a significant asymmetry was still found between their Spanish and English p-stranding. This is a bit surprising if we understand these bilinguals to have transferred English-like p-stranding to their Spanish. If their grammars have aligned for this construction, why are they not more uniformly aligned? To investigate this further, we can modify the way AoA of English is operationalized. Instead of dividing the participants into two separate groups with a specific age as a cut-off point, we can plot each participant's mean z-score for each language condition by their self-reported AoA of English[9]. This configuration of the results is presented in Figure 3. Here we can see that the same overall pattern holds. Note that p-stranding in English and in English-to-Spanish CS received stable, higher acceptability scores, regardless of the specific AoA of English of the participants. Meanwhile, p-stranding in both Spanish and Spanish-to-English CS showed a steady decline in acceptability as the AoA of English increased. When analyzed this way, we see that there was not a categorical divide regarding AoA, but rather a continuum. This point should not be surprising for researchers in bilingualism, but it is important to reiterate that although terms like *simultaneous* and *sequential* are helpful in discussing important differences in bilingual grammars, we need to constantly be mindful of heterogeneity within these groups.

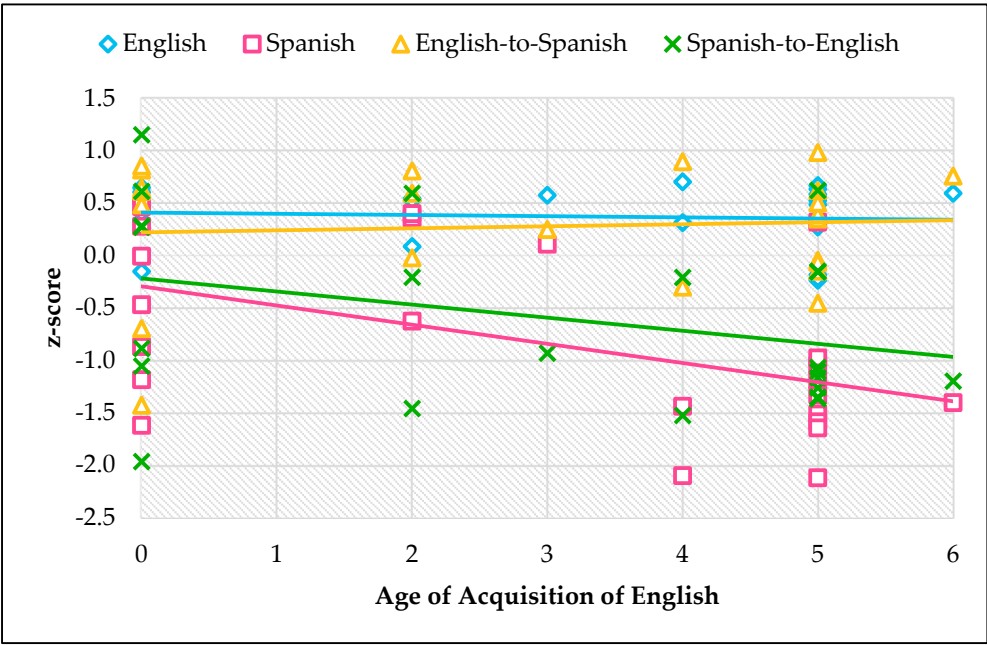

**Figure 3.** Average z-score by language(s) and AoA of English.

Overall, the results show that the availability of p-stranding in Spanish–English CS follows directly from whether a Spanish or English preposition can be stranded monolingually by a bilingual speaker. That is to say, the language of the preposition indicates whether p-stranding is available or not. Specifically, English prepositions allow p-stranding for any monolingual or mixed sentence regardless of the type of heritage speaker, while Spanish prepositions allow for it only if the speaker is a simultaneous bilingual.

## 5. Discussion

When it comes to the availability of p-stranding in Spanish–English intrasentential CS, the current study shows that it is possible, but both AoA of English and switch direction can mitigate its acceptability. As argued by Pascual y Cabo and Soler (2015), for heritage speakers of Spanish, p-stranding "is a domain of grammar that can be affected developmentally depending on the (socio)linguistic realities in which these individuals are immersed during the first years of acquisition" (p. 203). Syntactically speaking, the results from this experiment provide evidence that the Spanish of simultaneous bilinguals is influenced

by English in that there is no D+P incorporation. This creates a parallelism between their Spanish grammar and their English grammar, which not only allows for free extraction of DPs from PPs in monolingual contexts but also permits p-stranding when mixing their two grammars, with no restriction on CS (regardless of the direction of the switch). In short, if a simultaneous Spanish–English bilingual wishes to strand a preposition, they can do so when using English, Spanish, or any mixture of the two.

This finding for the simultaneous bilinguals raises an interesting issue. If it is indeed the case that there is no D+P incorporation for such individuals, this would suggest that there would never be any pied-piping with prepositions for them. Since the current study only tested p-stranding, we cannot make concrete claims as to whether the simultaneous biliguals do or do not optionally employ pied-piping in English, Spanish, and/or code-switching instead of p-stranding. Moreover, Law's (2006) proposal does not account for the optionality of pied-piping in English. Unlike in languages like Spanish where pied-piping is derived from the combination of wh-movement and the D+P incorporation, it is unclear what motivates optional pied-piping in English. Nevertheless, whatever accounts for it in monolingual speakers can be extended to the simultaneous bilinguals. That is to say, it may be the case that simultaneous bilinguals have drawn on some sort of optionality for p-stranding, as in English, based on the interplay of properties of D and P for English and Spanish.

Contrasting with the simultaneous bilinguals, the child sequential bilinguals show an asymmetry that is in line with what the literature has shown regarding monolingual speakers. Essentially, when it comes to p-stranding, their Spanish grammar is given time to develop the crucial syntactic restriction (i.e., D+P incorporation) before there is enough influence from English.[10] Interestingly, this asymmetry has specific consequences when it comes to mixing the two grammars in the same sentence. For child sequential bilinguals, extraction of a Spanish DP out of an English PP is acceptable, but not vice versa. This asymmetry is evidence that the D+P incorporation in their Spanish grammar presents itself in CS, but only in specific contexts. Recall that it was hypothesized that there would be a restriction on p-stranding in CS for child sequential bilinguals, but that it would be one of three different possibilities depending on the specific element(s) that initiates the D+P incorporation. Since p-stranding is available in Spanish-to-English switches and only in such switches, these results suggest that it is the language of the preposition and not the determiner that determines incorporation.

A lingering question is the extensiveness of the lack of D+P incorporation in Spanish for the simultaneous bilinguals. Recall that the availability of p-stranding in both CS contexts points to the absence of a unified syntactic unit, which is essential to Law's (2006) argument for how p-stranding is prohibited in languages like Spanish. However, also recall that the only preposition tested here was *with/con*. This form does not have any suppletive forms in Spanish, so all we can say for certain is that there is evidence that D+P incorporation does not occur with *con* 'with' for simultaneous Spanish–English bilinguals. It seems implausible to argue that there is no D+P incorporation at all for these speakers; although it was not tested here, we can presume that these speakers do employ the two Spanish suppletive forms, *del* 'of the' and *al* 'at the'. Assuming they do, future research should test whether p-stranding with the prepositions *de* 'of' and *a* 'to' behave the same way as *con* 'with' for heritage speakers of Spanish, or if there is perhaps variability depending on the particular preposition. It is possible that there is only a syntax-morphology-interface condition for *de* 'of' and *a* 'to', whereas all other Spanish prepositions (which lack any suppletive forms) are free to be stranded for these bilinguals.

Although the primary focus of this study is CS, it is important to explicitly acknowledge that the results provide additional syntactic evidence for the behavior of p-stranding in both Spanish and English. First, the current results replicate those of Pascual y Cabo and Soler (2015) for heritage speakers of Spanish, as a parallel difference was found between simultaneous and child sequential bilinguals. Additionally, the current results also provide more evidence of the general, ubiquitous availability of p-stranding in English.

## 6. Limitations and Conclusions

There are various limitations to the current study. As described earlier, the experiment only included one specific preposition, *with/con*, and as such we should not assume that the results here are representative of the way all prepositions behave regarding p-stranding in Spanish–English CS. There is known idiosyncratic variation with p-stranding depending on the preposition (Biber et al. 1999), so it should be expected that this would carry over to CS as well. Additionally, the data were obtained from an AJT, so it is unclear if production data would show the same patterns. It is possible that some participants are hesitant to accept p-stranding when providing a judgment, but then actually use p-stranding in their everyday speech without an issue (or vice versa). Finally, as already mentioned, heritage language speakers (like bilinguals more generally) are a heterogeneous group regarding their linguistic backgrounds, and here we only looked closely at their AoA of English. Continued investigation of other linguistic variables would likely provide more detailed information about the behavior of p-stranding in Spanish–English CS.

Nonetheless, the current study provides a clear first step toward understanding when p-stranding is available in CS. Since simultaneous Spanish–English bilinguals' grammars allow p-stranding in both languages, there is no restriction in CS. These results suggest there is no D+P incorporation in their grammars, allowing for free extraction of DPs from PPs. However, the asymmetry found for child sequential bilinguals shows that extraction of a Spanish DP out of an English PP is acceptable, but not vice versa. These findings are evidence that they have D+P incorporation in Spanish, which presents itself in switched contexts sometimes. Specifically, these results suggest that it is the preposition and not the determiner that determines incorporation, and as such the language of the preposition dictates whether there is p-stranding or not.

**Funding:** This research received no external funding.

**Institutional Review Board Statement:** The study was conducted according to the guidelines of the Declaration of Helsinki and approved by the Institutional Review Board of the University of Alabama (Protocol ID 18-03-1030, 21 June 2018).

**Informed Consent Statement:** Informed consent was obtained from all subjects involved in the study.

**Data Availability Statement:** The data presented in this study are openly available in IRIS at https://www.iris-database.org.

**Acknowledgments:** Special thanks to undergraduate researcher Rolf Tilley for his assistance in preparing the experimental materials. Additional thanks to the various instructors who helped in recruiting participants for the broader research project: Rafael Álvarez, Alicia Cipria, Mandy Faretta-Stutenberg, Ali Gonzenbach Perkins, Xabi Granja, Jessica Hubickey, Connie Janiga-Perkins, Ernesto Kortright, Marie-Eve Monette, Erin O'Rourke, Shirin Posner, Iñaki Rodeño, Laura Rojas-Arce, and Ana Skelton. Thanks as well to the College Academy of Research, Scholarship, and Creativity Activity (CARSCA) at the University of Alabama for providing funding for participant payment.

**Conflicts of Interest:** The author declares no conflict of interest.

## Appendix A

*Target English stimuli*

Bill doesn't know what woman Megan is arguing with.
Chad doesn't know what friend Kevin is traveling with.
Emma doesn't know what guy Jen is eating with.
Zoey doesn't know what classmate Josh is studying with.
Dave is the coach that Rob practices with.
Logan is the boy that Ashley lives with.
Lucy is the girl that Gabe is going out with.
Sally is the lady that Amy works with.

*Target Spanish stimuli*

> Araceli no sabe qué hombre Rosario está comiendo con.
> Fernando no sabe qué amiga Sergio está viajando con.
> Francisca no sabe qué compañero de clase Octavio está estudiando con.
> Manuel no sabe qué señora Ximena está discutiendo con.
> Carlos es el entrenador que Miguel practica con.
> Javier es el chico que Alejandra vive con.
> Leticia es la chica que Arturo está saliendo con.
> Yolanda es la mujer que Beatriz trabaja con.

*Target English-to-Spanish CS stimuli*

> Bill doesn't know what woman Ximena está discutiendo con.
> Chad doesn't know what friend Sergio está viajando con.
> Emma doesn't know what guy Rosario está comiendo con.
> Zoey doesn't know what classmate Octavio está estudiando con.
> Dave is the coach que Miguel practica con.
> Logan is the boy que Alejandra vive con.
> Lucy is the girl que Arturo está saliendo con.
> Sally is the lady que Beatriz trabaja con.

*Target Spanish-to-English CS stimuli*

> Araceli no sabe qué hombre Jen is eating with.
> Fernando no sabe qué amiga Kevin is traveling with.
> Francisca no sabe qué compañero de clase Josh is studying with.
> Manuel no sabe qué señora Megan is arguing with.
> Carlos es el entrenador that Rob practices with.
> Javier es el chico that Ashley lives with.
> Leticia es la chica that Gabe is going out with.
> Yolanda es la mujer that Amy works with.

*Filler English stimuli*

> Hannah always speaks English.
> Hannah speaks always English.
> Henry carefully reads instructions.
> Henry reads carefully instructions.
> Nate completely understands the homework.
> Nate understands completely the homework.
> Sarah buys frequently tomatoes.
> Sarah frequently buys tomatoes.
> Her friends have bought new shoes recently.
> His brothers have eaten pizza every day.
> The students have paid attention to the professor today.
> Your neighbors have visited that restaurant several times.
> He has a red big balloon.
> The earthquake destroyed the ancient beautiful city.
> The man ate a tuna delicious sandwich.
> What did you believe that has not died yet?
> Who did she say that speaks Spanish?
> Who did you think that is fishing?
> A minute ago that guy ordered a beer.
> A minute ago he ordered a beer.
> A minute ago you and him ordered a beer.
> Five minutes ago they started to dance.
> Five minutes ago those guys started to dance.
> Five minutes ago you and them started to dance.

Last week he met our grandmother.
Last week that guy met our grandmother.
Last week you and him met our grandmother.
Yesterday they bought some peaches.
Yesterday those guys bought some peaches.
Yesterday you and them bought some peaches.
His sister eats more eggs than he every morning.
His sister eats more eggs than him every morning.
His sister eats more eggs than that guy every morning.
My friend drinks more wine than he every night.
My friend drinks more wine than him every night.
My friend drinks more wine than that guy every night.
Our employees read more books than they every month.
Our employees read more books than those guys every month.
Our employees read more books them every month.
Your colleagues work more hours than them every week.
Your colleagues work more hours than they every week.
Your colleagues work more hours than those guys every week.

*Filler Spanish stimuli*

Antonio cuidadosamente lee las instrucciones.
Antonio lee cuidadosamente las instrucciones.
José completamente entiende la tarea.
José entiende completamente la tarea.
Juana habla siempre español.
Juana siempre habla español.
Margarita compra frecuentemente tomates.
Margarita frecuentemente compra tomates.
Los estudiantes han prestado atención al profesor hoy.
Sus amigas han comprado zapatos nuevos recientemente.
Sus hermanos han comido pizza todos los días.
Tus vecinos han visitado ese restaurante varias veces.
David y Diego han ya pedido cinco dólares.
El mapa cuesta nada.
Ellos han siempre tenido muchas actividades diferentes.
Ellos les han brevemente dado una oportunidad de salir.
José Miguel tiene ningún cuchillo.
La gente visita el museo nunca.
Ayer ellos compraron unos duraznos.
Ayer esos hombres compraron unos duraznos.
Ayer tú y ellos compraron unos duraznos.
Hace cinco minutos ellos empezaron a bailar.
Hace cinco minutos esos hombres empezaron a bailar.
Hace cinco minutos tú y ellos empezaron a bailar.
Hace un minuto él pidió una cerveza.
Hace un minuto ese hombre pidió una cerveza.
Hace un minuto tú y él pidieron una cerveza.
La semana pasada él conoció a nuestra abuela.
La semana pasada ese hombre conoció a nuestra abuela.
La semana pasada tú y él conocieron a nuestra abuela.
Mi amigo bebe más vino que él todas las noches.
Mi amigo bebe más vino que ese hombre todas las noches.
Nuestros empleados leen más libros que ellos cada mes.
Nuestros empleados leen más libros que esos hombres cada mes.
Su hermana come más huevos que él cada mañana.

Su hermana come más huevos que ese hombre cada mañana.
Sus colegas trabajan más horas que ellos cada semana.
Sus colegas trabajan más horas que esos hombres cada semana.

*Filler English-to-Spanish CS stimuli*

Hannah always habla español.
Hannah speaks siempre español.
Henry carefully lee las instrucciones.
Henry reads cuidadosamente las instrucciones.
Nate completely entiende la tarea.
Nate understands completamente la tarea.
Sarah buys frecuentemente tomates.
Sarah frequently compra tomates.
Her friends han comprado zapatos nuevos recientemente.
Her friends have comprado zapatos nuevos recientemente.
His brothers han comido pizza todos los días.
His brothers have comido pizza todos los días.
The students han prestado atención al profesor hoy.
The students have prestado atención al profesor hoy.
Your neighbors han visitado ese restaurante varias veces.
Your neighbors have visitado ese restaurante varias veces.
Everyone will get wet si llueve hoy.
I will leave si me siento mal.
She hides cuando él la llama.
Sometimes he'll go to the store y olvida lo que estaba buscando.
We'll hear a sound si alguien toca el timbre.
We'll tell him si lo vemos.
A minute ago that guy pidió una cerveza.
A minute ago he pidió una cerveza.
A minute ago you and him pidieron una cerveza.
Five minutes ago they empezaron a bailar.
Five minutes ago those guys empezaron a bailar.
Five minutes ago you and them empezaron a bailar.
Last week he conoció a nuestra abuela.
Last week that guy conoció a nuestra abuela.
Last week you and him conocieron a nuestra abuela.
Yesterday they compraron unos duraznos.
Yesterday those guys compraron unos duraznos.
Yesterday you and them compraron unos duraznos.
His sister eats more eggs than él cada mañana.
His sister eats more eggs than ese hombre cada mañana.
My friend drinks more wine than él todas las noches.
My friend drinks more wine than ese hombre todas las noches.
Our employees read more books than ellos cada mes.
Our employees read more books than esos hombres cada mes.
Your colleagues work more hours than ellos cada semana.
Your colleagues work more hours than esos hombres cada semana.

*Filler Spanish-to-English CS stimuli*

Antonio cuidadosamente reads instructions.
Antonio lee carefully instructions.
José completamente understands the homework.
José entiende completely the homework.
Juana habla always English.
Juana siempre speaks English.

Margarita compra frequently tomatoes.
Margarita frecuentemente buys tomatoes.
Los estudiantes han paid attention to the professor today.
Los estudiantes have paid attention to the professor today.
Sus amigas han bought new shoes recently.
Sus amigas have bought new shoes recently.
Sus hermanos han eaten pizza every day.
Sus hermanos have eaten pizza every day.
Tus vecinos han visited that restaurant several times.
Tus vecinos have visited that restaurant several times.
A veces va a la tienda and forgets what he was looking for.
Ella se esconde when he calls her.
Le diremos if we see him.
Todos van a mojarse if it rains today.
Vamos a escuchar un sonido if someone rings the doorbell.
Voy a salir if I feel sick.
Ayer ellos bought some peaches.
Ayer esos hombres bought some peaches.
Ayer tú y ellos bought some peaches.
Hace cinco minutos ellos started to dance.
Hace cinco minutos esos hombres started to dance.
Hace cinco minutos tú y ellos started to dance.
Hace un minuto él ordered a beer.
Hace un minuto ese hombre ordered a beer.
Hace un minuto tú y él ordered a beer.
La semana pasada él met our grandmother.
La semana pasada ese hombre met our grandmother.
La semana pasada tú y él met our grandmother.
Mi amigo bebe más vino que he every night.
Mi amigo bebe más vino que him every night.
Mi amigo bebe más vino que that guy every night.
Nuestros empleados leen más libros que them every month.
Nuestros empleados leen más libros que they every month.
Nuestros empleados leen más libros que those guys every month.
Su hermana come más huevos que he every morning.
Su hermana come más huevos que him every morning.
Su hermana come más huevos que that guy every morning.
Sus colegas trabajan más horas que them every week.
Sus colegas trabajan más horas que they every week.
Sus colegas trabajan más horas que those guys every week.

## Notes

[1]   This ungrammaticality is not universal for all speakers of Spanish, as will be discussed in more detail in Section 2.

[2]   The two languages do align in that English optionally allows for pied-piping instead of p-stranding, which is addressed later in Section 2.

[3]   As is customary in CS research, italics are used for one language and not the other to easily denote switching. Throughout this document, all CS examples will have Spanish italicized and English not. In the actual experiment, though, all sentences (monolingual and switched) were presented to the participants without any typographical distinction between the two languages.

[4]   A "dehydrated" sentence is one that consists of separated segments (in this case, by forward slashes) that do not form a complete sentence. Participants are asked to rewrite those segments as a full sentence, making modifications as necessary, with the idea being that they will correct the structures to model their grammar. For example, *Xabi/gustar/chocolate* 'Xabi/to like/chocolate' would produce something like *A Xabi le gusta el chocolate* 'Xabi likes chocolate.'

[5]   Regarding choice of modality, see Koronkiewicz and Ebert (2018) for evidence showing that written and oral stimuli produce comparable results in CS AJT research.

6    Although the focus of this study is on CS, see Ebert and Koronkiewicz (2018) for more details about the necessity of including monolingual judgments in an AJT-based CS study.

7    The adverb stimuli were included as part of a separate study on verb raising. The auxiliary verb construction was chosen as part of the filler stimuli because a switch between the present perfect auxiliary verb and its participle as well as has been consistently shown in the literature to be ungrammatical for Spanish–English bilinguals. For an overview of auxiliary switches, see Guzzardo Tamargo (2012). The pronoun construction was chosen similarly because a switch between a subject pronoun and a finite verb has been consistently shown in the literature to be ungrammatical for Spanish–English bilinguals. For an overview of pronoun switches, see Koronkiewicz (2014) and González-Vilbazo and Koronkiewicz (2016).

8    No significant difference was found between the z-scores for the embedded wh-movement and relative clause stimuli, so the results presented here do not differentiate between the two syntactic constructions.

9    The exact wording from the background questionnaire was: "At what age did you start acquiring English? Enter 0 to indicate since birth.".

10    As an anonymous reviewer noted, we could we explain this in terms of reduced amount of input in Spanish compared to the amount of input the child sequential bilinguals received.

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
