# Peer review of "Preposition Stranding in Spanish–English Code-Switching"

_languages, doi:10.3390/languages7010045_

Round 1

Reviewer 1 Report

ReviewPREPOSITIONAL STRANDING IN SPANISH-ENGLISH CODE-SWITCHING”

 Summary: The following study investigates how simultaneous and sequential bilinguals with the language combination Spanish-English accept, via an acceptability judgement task, code-switched sentences with prepositional stranding. This phenomenon is only available in English, whereas both English and Spanish share the pied-piping structure, in which the DP, together with the preposition, are moved to a higher position in the clause. The results of this written task show, as already previous studies with heritage speakers of Spanish in the US for monolingual utterances have shown, that sequential bilinguals accept p-stranding in English but reject it for Spanish. The CS items mirror the monolingual results: those CS utterances with an English preposition and a Spanish DP are accepted as p-stranding structures, while p-stranding is rejected for Spanish prepositions and English DPs. By contrast, the results for the simultaneous bilingual group with monolingual utterances go in line with previous studies. Interestingly, when taking the CS results in consideration, this group also accepts p-stranding with Spanish preposition (and an English DP) to a certain extend. 

General comments: The results of this empirical study are really interesting for the empirical findings on CS from different perspectives. It mirrors the results from Cabo y Pascual & Soler (2015) to p-stranding in English and Spanish by bilingual speakers with different age of acquisition profiles (simultaneous and sequentials) and expands these results with a new acceptability judgement task with CS utterances containing the preposition with/con followed by a English/Spanish DP. Briefly, I would recommend the author(s) to expand and be more precise on the current literature and discussion for the term heritage speaker as well as on age of acquisition/age of onset for bilingualism (see the specific comments in the manuscript for more details). Moreover, I would like to suggest that the author(s) move the results on age of acquisition for the simultaneous bilingual group from the discussion to the results section, since these clearly show how age of exposure to English (whether from birth or at age 1;0/2;0/3;0/etc.) diminshes the acceptability of p-stranding in Spanish and in CS utterances containing a Spanish preposition and an English DP.

Specific comments: See notes in the manuscript

Author Response

Please see attached document for the response to reviewers' comments.

Reviewer 2 Report

Preposition stranding in Spanish-English code-switching

The article shows the result that simultaneous bilinguals accept preposition stranding in both of their languages, English as well as Spanish, in monolingual and code-switched constructions (Spanish-to-English as well as English-to-Spanish). Sequential bilinguals showed a (expected) difference between their two languages, again both in monolingual and code-switched constructions. The method is a grammaticality judgement task.

1). With respect to the presentation of p-stranding in the two languages, Spanish and English, I asked myself whether the distinction outlined is really valid for all languages (and all prepositions in the presented languages). In a language with determiner-incorporation, like French, there is p-stranding with some prepositions, like pour ‘for’ and contre ‘against’: le president que j’ai vote pour ‘the president that I voted for’. French has incorporated determiners, like the sequence “à” (prep) and “le” (definite masculine determiner) which is obligatorily expressed as “au”. And in the languages where D does not incorporate into P, like in English, is p-stranding really possible with all prepositions? Since the task tested CON and WITH, could it be the case that these preps are lexically marked with this option (p-stranding) in many languages? Take French again. Here, the following constructions are perfect

Les amis que je sors avec (the friends that I go-out with)

Un document que je ne voyage pas sans (a document that I NOT travel NOT without)

L’église que je passe devant (the church that I pass before)

(cf. Jones, M. 1996, Foundations of French syntax, p. 517)

Please check with native speakers of Spanish whether these constructions are possible in Spanish as well. Jones (1996: 518) argues that the French examples are not instances of p-stranding, but that there is a resumptive pronoun in these constructions. Although PPs, accordingly, must be analyzed as islands in French, learners of French do have these constructions in their input. It is therefore possible that a simultaneous bilingual has analyzed these constructions (if they exist in native Spanish as well) with the grammatical analysis of English, thus giving rise to the results in figure 2.

But there may be another explanation: Could it also be the case that the simultaneous bilinguals have similar judgements in both of their languages because that’s what bilinguals do in this type of test? Is there psycholinguistic evidence in favor of the assumption that simultaneous bilinguals are like sequential bilinguals in production tasks, they clearly separate their two languages (we know that they do!), but that grammaticality judgement tasks give rise to a behavior which makes Spanish a language in which p-stranding is accepted? What is the role of inhibition of the other language in relation to the task? Perhaps, Grosjean (1998) is of help here.

2) What is meant by the term “requisite” in “requisite pied-piping”?

3) Describe Law’s approach in more detail. It is unclear how one instance of D-incorporation into P should be responsible for the language as a whole to not accept p-stranding. Is Law’s approach feature-driven?

4) It would be extremely important to discuss the results in the framework of incomplete acquisition, parameter setting (in relation to the grammatical domain in question, p-stranding) and the issue of cross-linguistic influence. I attach some articles which may be of help (although the possibility that the task as such was responsible for the results in the simultaneous bilinguals should may change the relevance of these issues): Tsimpli (2014), Schulz & Grimm (2019), Arnaus Gil, Stahnke & Müller (2021). As for cross-linguistic influence in the simultaneous bilinguals, the authors may look at: Hulk & Müller (2000), Müller & Hulk (2001).

5) aural=oral

6) Justify why age 5 was the turning point.

7) In (18) indicate why * has been chosen. Which was the constraint (Functional Head Constraint) which makes the construction ungrammatical?

8) Is it not exactly clear to me what the relevance of the code-switched sentences is. Why would a switch like in (18a) be ungrammatical if “nothing constrains code-switching apart from the two grammars involved”? Are the results on CS presented in order to show that non-switched and switched sentences are treated alike if it comes to (the judgement of the grammaticality of) p-stranding? Please explain further and outline further why this is relevant. How could one explain that the sequential bilinguals differed with respect to their judgements of monolingual Spanish sentences and switched sentences from English-to-Spanish?

9) Figure 2: Explain the y-axis, what does it mean if one has reached -1.50? the acceptability is low, right, but what is meant by 1.50?

10) Depending on the answer to 9): Does it mean that in English, all p-stranding constructions were accepted by the simultaneous and by the sequential bilinguals? If not, why so? And what about Spanish? Have all p-stranding constructions been rejected (at least the non-switched) by the sequential bilinguals? If not, why so?

11) Discussion: Do you really mean that the Spanish of the sequential bilinguals is being influenced? (line 345)

12) I believe it would be clearer to say that it is the LANGUAGE of the preposition which gives rise to whether p-stranding is judged as grammatical or not (less so).

13) I am a little lost here. How can the language of the determiner determine p-stranding? This was neither tested, as far as I understood, nor would it be relevant in the switched constructions, right? As far as I understood Law’s analysis of p-stranding from the text, once the language has incorporation of D into P, like Spanish, p-stranding is not an option in grammar. Why then test the determiner in CS?

14) I had difficulties to understand what is meant by the sentence at line 409-412. What does it mean to say: general tendency of how early exposure occurs?

15) Line 438: This is true, but determiners were not tested.

16) It is possible that the work by Klein (1995) helps a little.

Author Response

(The authors gave the same response as above.)

Reviewer 3 Report

Review

Review of “Preposition stranding in Spanish-English code-switching”

Recommendation

The paper contributes to our descriptive understanding of interesting phenomena around p-stranding in English and Spanish CS. It should be published with some revisions, suggested below.

Comments

I am not sure the participants are heritage language speakers in the usual sense. A heritage language is a minoritized language which children learn from members of their family at home, but generally under conditions of limited input. See Polinsky and Kagan (2007) for discussion. Valdés (2000), specifically, refers to heritage speakers as “individuals raised in homes where a language other than English is spoken and who are to some degree bilingual in English and the heritage language.”  By contrast, the participants in the present study seem to be second language learners of English (sequential bilinguals) and simultaneous bilinguals who grew up speaking both English and Spanish. We are not offered enough biographical detail to know if they are heritage speakers; it appears that only onset of age of acquisition of English, a majority language, is known.  Although the special issue of the journal is concerned with heritage language speakers, the author may need to dispense with using that term in the present submission as it is not clearly relevant.

P5, line 175. The authors cite Grimstad et al. (2018) and MacSwan (1999) as foundational for the constraint-free approach to the grammar of CS which the paper adopts, but Grimstad et al. (2018) is not foundational or specifically relevant to that proposal. So it is not clear why the author cites Grimstad et al. (2018) in that specific connection, and Grimstad et al. is not subsequently used in the piece.  While the generative implementation of the constraint-free approach is original to MacSwan (1999), it is best elucidated in the introductory chapter to MacSwan’s (2014) MIT Press volume. The author should review that specific reference and consider using the term “constraint-free approach” to name the specific theoretical orientation adopted in the paper.  MacSwan (2021) also reviews these issues. More detailed description of the CS literature may be appropriate.

The acceptability pattern for simultaneous bilinguals for the CS conditions (*English-to-Spanish, with Spanish-to-English ok) makes sense if the preposition is the trigger for incorporation, as the author says, but the results for the simultaneous bilinguals, who show more tolerance for the English-to-Spanish condition is surprising. To me, the author’s conjectures about this is not persuasive. It’s concluded (page 10) that “Syntactically speaking, the results from this experiment provide evidence that the Spanish of sequential bilinguals is influenced by English in that there is no D+P incorporation.” But if there was no D+P incorporation, they would never pied-pipe prepositions. Presumably pied-piping is required for Spanish Ps and optional in some way for English Ps.  How, for instance, does Law (whom the author relies for p-stranding theory) account for the optionality of pied-piping in English? And why would simultaneous bilinguals somehow assimilate their English and Spanish grammars but not sequential bilinguals?

It may be the case that simultaneous speakers have drawn on some sort of optionality for p-stranding, as in English, based on the interplay of properties of D and P for English and Spanish, while the sequentials are treating p-standing as non-optional when the proposition is Spanish, whether D is English or Spanish.

I suggest that the author provides a more detailed account of the syntactic mechanisms underlying P-D movement, from Law (2006) and subsequent work (like Law’s more recent handbook chapter). A crucial question is how P movement takes place in the case of English where it is seen as optional, but somewhat contrived (to my ear). And I presume there are languages where P movement is not permitted, and stranding is required.  This detailed analytical context will provide better background for making inferences about why the simultaneous bilinguals are more tolerant of p-stranding in both CS conditions, and perhaps why sequential bilinguals – whose English can reasonably be said to be conditioned by Spanish, but not vice versa.

In any case, the treatment of the simultaneous bilinguals’ Spanish as somehow influenced or conditioned by their English grammar does not seem reasonable to me.

References

MacSwan, J. (2014). Programs and proposals in codeswitching research:  Unconstraining theories of bilingual language mixing, pp. 1-33. In J. MacSwan (ed.), Grammatical Theory and Bilingual Codeswitching. Cambridge: MIT Press.

MacSwan, J. (2021). Theoretical approaches to the grammar of codeswitching, pp. 88-109. In E. Adamou & Y. Matras (eds.), Routledge Handbook of Language Contact.  New York: Routledge.

Polinsky, M. & Kagan, O. (2007). Heritage Languages: In the 'Wild' and in the Classroom. Language and Linguistics Compass. 1. 10.1111/j.1749-818X.2007.00022.x.

Valdés, G. (2000). The teaching of heritage languages: An introduction for Slavic-teaching professionals, ed. by Olga Kagan and Benjamin Rifkin.  The Learning and Teaching of Slavic Languages and Cultures, 375-403. Bloomington, IN.: Slavica.

Author Response

(The authors gave the same response as above.)

Round 2

Reviewer 1 Report

The author has taken all comments into consideration and has applied them accordingly to the manuscript. I would suggest to publish the manuscript in its current version.

Just one small typing error in line 143: "been" instead of "be".

Reviewer 2 Report

Thank you for considering my comments. The study is very interesting and will surely receive attention from acquisitionists. Figure 3 is clear now and extremely interesting! 

footnote 7: There is something wrong with "as well as"

Footnote 10: "we could we"

Page 13: "... can BE extended"